# The Receptor Tyrosine Kinase TrkA Is Increased and Targetable in HER2-Positive Breast Cancer

**DOI:** 10.3390/biom10091329

**Published:** 2020-09-17

**Authors:** Nathan Griffin, Mark Marsland, Severine Roselli, Christopher Oldmeadow, John Attia, Marjorie M. Walker, Hubert Hondermarck, Sam Faulkner

**Affiliations:** 1School of Biomedical Sciences and Pharmacy, Faculty of Health and Medicine, University of Newcastle, Callaghan, NSW 2308, Australia; nathan.griffin@uon.edu.au (N.G.); mark.marsland@newcastle.edu.au (M.M.); severine.roselli@newcastle.edu.au (S.R.); sam.faulkner@newcastle.edu.au (S.F.); 2Hunter Medical Research Institute, University of Newcastle, New Lambton Heights, NSW 2305, Australia; christopher.oldmeadow@newcastle.edu.au (C.O.); john.attia@newcastle.edu.au (J.A.); marjorie.walker@newcastle.edu.au (M.M.W.); 3School of Mathematical and Physical Sciences, Faculty of Science and Information Technology, University of Newcastle, Callaghan, NSW 2308, Australia; 4School of Medicine and Public Health, Faculty of Health and Medicine, University of Newcastle, Callaghan, NSW 2308, Australia

**Keywords:** breast cancer, tyrosine kinase receptor A (NTRK1/TrkA), human epidermal growth factor receptor 2 (HER2), clinical biomarker, therapeutic target

## Abstract

The tyrosine kinase receptor A (NTRK1/TrkA) is increasingly regarded as a therapeutic target in oncology. In breast cancer, TrkA contributes to metastasis but the clinicopathological significance remains unclear. In this study, TrkA expression was assessed via immunohistochemistry of 158 invasive ductal carcinomas (IDC), 158 invasive lobular carcinomas (ILC) and 50 ductal carcinomas in situ (DCIS). TrkA was expressed in cancer epithelial and myoepithelial cells, with higher levels of TrkA positively associated with IDC (39% of cases) (*p* < 0.0001). Interestingly, TrkA was significantly increased in tumours expressing the human epidermal growth factor receptor-2 (HER2), with expression in 49% of HER2-positive compared to 25% of HER2-negative tumours (*p* = 0.0027). A panel of breast cancer cells were used to confirm TrkA protein expression, demonstrating higher levels of TrkA (total and phosphorylated) in HER2-positive cell lines. Functional investigations using four different HER2-positive breast cancer cell lines indicated that the Trk tyrosine kinase inhibitor GNF-5837 reduced cell viability, through decreased phospho-TrkA (Tyr490) and downstream AKT (Ser473) activation, but did not display synergy with Herceptin. Overall, these data highlight a relationship between the tyrosine kinase receptors TrkA and HER2 and suggest the potential of TrkA as a novel or adjunct target for HER2-positive breast tumours.

## 1. Introduction

Targeted therapies against the human epidermal growth factor receptor-2 (HER2) have demonstrated their clinical efficacy in breast cancer. However, approximately 50% of HER2 overexpressing breast tumours are refractory to current anti-HER2 approaches [1]. HER2 positivity represents 15–20% of all breast cancers and is defined by protein expression, HER2 gene copy number alterations or by the ratio of HER2/CEP17 [2,3]. Unfortunately, there are no clinically validated predictive biomarkers and the use of anti-HER2 agents is still determined based on the HER2 status of the primary tumour. However, a key oversight in current diagnostic and prognostic methodologies is that HER2 status of the primary tumour may vary compared to metastases in approximately 25% of all cases [4,5]. This further highlights the crucial need to identify biomarkers of resistance and metastatic progression as well as adjunct therapeutic targets for improving the management of HER2-positive breast cancer [2,6].

The tyrosine kinase receptor TrkA (or neurotrophin receptor kinase 1, NTRK1) has been predominately studied for its role during the development of the nervous system. In developing neurons, TrkA activation upon binding of its ligands nerve growth factor (NGF) or neurotrophin-3 (NT-3) results in the stimulation of various tyrosine kinase-induced signaling pathways leading to neuronal outgrowth [7]. In the adult, where neurogenesis is largely reduced, TrkA acts as a pain receptor in sensory neurons and contributes to the transmission of pain signals to the central nervous system [8]. Interestingly, TrkA is also oncogenic and there is accumulating evidence for its overexpression and involvement in cancer progression [9,10,11,12,13,14], including in the form of TrkA fusion proteins as described in lung cancer [15]. Furthermore, TrkA is a therapeutic target in oncology and Trk inhibitors are being tested in several clinical trials [16,17,18,19]. In breast cancer, TrkA has been shown to stimulate tumour cell invasion and metastasis through the activation of ERK, PI3K, SRC and AKT signaling pathways [10,20,21,22]. TrkA (*NTRK1*) gene overexpression has also been shown to occur preferentially in invasive tumours [23]. In breast cancer patients, increased TrkA phosphorylation has been described in malignant pleural effusions [24] while phosphorylation of Tyr674/675 and activation of tyrosine kinase activity in the cytoplasmic domain of TrkA has been shown as a potential prognostic biomarker [25]. However, TrkA protein expression in a large clinically relevant cohort of breast tumours, as well as its overall clinical significance remain unclear [26].

The present study aimed to elucidate the immunohistochemical profile and clinical ramifications of TrkA expression in a large and pathologically diverse cohort of breast cancers. We report a significantly higher protein expression of TrkA in invasive ductal carcinomas (IDC) as compared with invasive lobular carcinomas (ILC) and ductal carcinomas in situ (DCIS). Interestingly, TrkA protein expression was also associated with HER2 status. Furthermore, in vitro experiments using a panel of breast cancer cell lines confirmed an overexpression of TrkA and phospho-TrkA in HER2-positive cell lines and demonstrated that targeting phospho-TrkA signaling, using the Trk inhibitor GNF-5837, resulted in decreased cell viability. Overall, these data suggest a potential utility of TrkA as an additional therapeutic target in HER2-positive breast cancers.

## 2. Materials and Methods

### 2.1. Breast Tissue Samples

High-density tissue microarrays (TMA, catalogue numbers; BR1921, BR1921a and BR8011) were obtained from US Biomax Inc. (Rockville, MD, USA) which included 316 cases of invasive breast carcinomas (158 IDC and 158 ILC) and 50 cases of non-invasive DCIS. The following clinicopathologic information was available for analysis: patient age, histopathological diagnosis, classification of malignant tumours (TNM), stage and grade. Estrogen receptor (ER), progesterone receptor (PR) and human epidermal growth factor receptor 2 (HER2) status was also provided. US Biomax Inc. quality controls are described as follows. Each single tissue spot on every array slide is individually examined by pathologists certified according to WHO published standardizations of diagnosis, classification and pathological grade. Each specimen collected was consented to by both hospital and individual. Discrete legal consent was obtained and the rights to hold research uses for any purpose or further commercialized uses were waived. This study was conducted in accordance with the Declaration of Helsinki, and the protocol approved by the Human Research Ethics Committee of the University of Newcastle, Australia (HREC reference: H-2012-0063).

### 2.2. Immunohistochemistry and Quantification of TrkA Expression

Immunohistochemistry was performed as previously described by our laboratory [27]. A TrkA antibody (1:200 dilution; catalogue number 2508, Cell Signaling Technology, Danvers, MA, USA) was applied to all TMAs. TMAs were digitized using the Aperio AT2 scanner (Leica Biosystems, Victoria, Australia) at 400× absolute resolution. Two independent observers (S.F. and N.G.) and an experienced pathologist (M.M.W.) quantified the intensity of TrkA staining as 0 (no staining), 1 (weak staining), 2 (moderate staining) and 3 (strong staining), as described previously [27,28]. Negative control testing was also performed using a rabbit (DA1E) monoclonal antibody IgG Isotype Control (0.8 μg/mL; catalogue number 3900, Cell Signaling Technology) and is shown in Appendix A. Clinicopathologic parameters were compared with TrkA staining intensity and analysed as ordinal variables using chi-squared (χ2) or Fisher’s exact tests. To test for statistically significant associations, we developed log-linear models to adjust for the bivariate associations for other potential confounders, which provided an additional chi-squared (χ2) test adjusted for all other variables; these included carcinoma type (IDC vs. ILC vs. DCIS), lymph node invasion (yes/no), ER+ (yes/no), PR+ (yes/no) and HER2+ (yes/no). Analyses were based on complete cases and performed using Stata (version 14.1, Statacorp, College Station, TX USA), with additional figures created using Prism (version 8.2.0, GraphPad Software, San Diego, CA, USA).

### 2.3. Cell Culture

Breast cancer cell lines MDA-MB-231 (catalogue number HTB-26), MCF-7 (catalogue number HTB-22), BT-474 (catalogue number HTB-20) and SK-BR-3 (catalogue number HTB-30) were purchased from the American Type Culture Collection (ATCC, Manassas, VA, USA). JIMT-1 cells were purchased from the Leibniz Institute DSMZ (catalogue number ACC 589, Braunschweig, Germany). The C-4 I cervical cancer cell line (TrkA-negative control) was purchased from the ATCC (catalogue number CRL-1594). ATCC and DSMZ routinely provide short tandem repeat (STR) verification of cell line authenticity upon purchase and receipt. Brain metastatic 231-BR cells and the human mammary epithelial (HME) cell line (transformed but non-tumourigenic) were a generous gift from Dr. Patricia S. Steeg (National Cancer Institute, Bethesda, MD, USA) and Dr. Robert A. Weinberg (Massachusetts Institute of Technology, Cambridge, MA, USA), respectively. The authenticity of 231-BR and HME cell lines were validated using the GenePrint 10 System (catalogue number B9510, Promega, Madison, WI, USA). All cell lines were maintained in DMEM (catalogue number 11960-044, Life Technologies Australia Pty. Ltd., Victoria, Australia) with 10% (*v/v*) foetal bovine serum (FBS, catalogue number SFBS-F, Bovogen Biologicals Pty Ltd., Victoria, Australia) and GlutaMAX™ Supplement (catalogue number 35050061, Life Technologies Australia Pty. Ltd.) and maintained in a humidified incubator (catalogue number CB-S 170, Binder, POCD Scientific, New South Wales, Australia) at 37 °C with 5% (*v/v*) CO_2_. Routine Mycoplasma testing was performed using the MycoAlert Mycoplasma Detection Kit (catalogue number LT07-118, Lonza, Basel, Switzerland). Cells were not maintained in culture for longer than 3 months to ensure passage number remained fit for purpose.

### 2.4. Protein Extraction and Western Blotting

Protein extraction from cell lines and Western blotting experiments were performed as previously described [29]. Complete mini-protease inhibitor cocktail tablets (catalogue number 4693124001, Roche, Basel, Switzerland) and PhosSTOP phosphatase inhibitor tablets (catalogue number 4906837001, Roche) were also used. Samples were centrifuged at 16,000× *g* for 15 min at 4 °C and the supernatant was extracted and stored at −80 °C prior to Western blotting. An anti-TrkA antibody (catalogue number ANT018, Alomone Labs, Jerusalem, Israel) was used at a dilution of 1:500 and a β-actin antibody (catalogue number A2066, Sigma-Aldrich) was used at a 1:5000 dilution as the equal loading control. The following antibodies from Cell Signaling Technology were all used at a dilution of 1:500 to assess cellular protein and downstream signaling pathways: anti-HER2/ErbB2 (catalogue number 2242), anti-phospho-TrkA (anti-p-TrkA) (Tyr490, catalogue number 9141), anti-p44/42 MAPK (Erk1/2) (catalogue number 9107), anti-phospho-p44/42 MAPK (p-Erk1/2) (Thr202/Tyr204, catalogue number 4370), anti-SRC (catalogue number 2123), anti-phospho-SRC (anti-p-SRC) (Tyr416, catalogue number 2101), anti-AKT (catalogue number 2920) and anti-phospho-AKT (anti-p-AKT) (Ser473, catalogue number 4060).

### 2.5. GNF-5837 Treatment and Cell Viability Assay

The antiproliferative efficacy of GNF-5837 was tested on HER2-positive breast cancer cell lines (SK-BR-3, BT-474, MDA-MB-453 and JIMT-1) using the CellTiter-Blue^®^ Cell Viability Assay (catalogue number G8080, Promega, WI, USA), as per the manufacturer’s instructions. Briefly, cells were seeded (5 × 10^3^ cells/well) into 96-well plates in 100 μL media containing 10% (*v/v*) FBS and incubated for 48 h at 37 °C with 5% (*v/v*) CO_2_. Cells (60% confluent) were then treated with GNF-5837 (catalogue number SML0844, Sigma-Aldrich, St. Louis, MO, USA) to a final concentration of 0–40 µM as well as 0.1% DMSO (catalogue number D2650, Sigma-Aldrich) as the negative control. At 48 h post-treatment, the media was replaced with fresh media containing 20% *v/v* CellTiter-Blue^®^ solution for an additional 4 h to assess cell viability. The fluorescent intensity of each well was then quantitated using a using the FLUOstar Optima Microplate Reader with a 560(20)_Ex_/590(10)_Em_ filter set (BMG LABTECH Pty. Ltd., Victoria, Australia) and the rate of cell viability was recorded and plotted as a percentage of the 0.1% DMSO negative control. The sensitivity of the HER2-positive breast cancer cell lines was calculated by half-maximal inhibitory concentration (IC_50_) estimated from the plot of the cell viability percentage versus increasing concentrations of the GNF-5837 administered, with data reported as the mean ± standard deviation. Each treatment was carried out using three biological replicates and experiments were repeated on three separate occasions. Statistical analysis was conducted with Prism (version 8.2.0, GraphPad Software) using an ordinary one-way analysis of variance (one-way ANOVA) test and the Tukey–Kramer post-test for assessing multiple comparisons.

## 3. Results

### 3.1. NTRK1 (TrkA) Gene Expression in Breast Cancer

Before investigating the protein level of TrkA using immunohistochemistry, we performed data mining of TrkA (*NTRK1*) gene expression, using the cBioPortal cancer genomics platform [30] of the PAM50 dataset from The Cancer Genome Atlas (TCGA) database [31,32]. The PAM50 dataset contains 825 samples of invasive breast carcinomas with stratification in terms of molecular subtypes of breast tumours. The results (Figure 1) indicated that *NTRK1* gene expression was altered in 6% (52 cases) of breast tumours with 31 amplifications, 3 missense mutations, 13 mRNA upregulations and 5 mRNA downregulations (Figure 1A). A total of 17% (14 cases) of basal (triple-negative) cancers had alterations within the *NTRK1* gene (Figure 1B). Luminal A and luminal B cancers presented with *NTRK1* alteration in 7% (Figure 1C) and 8% (Figure 1D) of cases, respectively. In HER2-enriched tumours, only 3% of cases presented with *NTRK1* alterations (Figure 1E). Overall patient survival, when considering all 825 cases of invasive breast cancers included in the PAM50 dataset, tended to be lower for cases with *NTRK1* gene alterations as compared to cases with no alterations (Figure 1F), but the statistical significance was limited (log-rank test *p*-value = 0.07). Alterations were grouped and included a combination of both up and down regulation of the mRNA transcript, however most alterations (90%) observed in the analyses were amplifications and/or mRNA upregulations. When considering each molecular subtype individually, no association between *NTRK1* gene expression and overall patient survival could be detected.

### 3.2. TrkA Is Overexpressed in HER2-Positive Breast Tumours

TrkA immunostaining in breast tumours (Figure 2) was observed preferentially in myoepithelial and cancer epithelial cells of IDC (Figure 2A–F), ILC (Figure 2G–I) and DCIS (Figure 2J–K). The intensity of TrkA labelling was quantified as 0 (no staining), 1 (weak intensity), 2 (moderate intensity) and 3 (strong intensity). The frequency distribution of TrkA labelling and association with patient clinicopathologic parameters is presented in Table 1. TrkA protein expression was higher in IDC than in ILC and DCIS. The proportion of IDC positive for TrkA was 39% whereas it was only 20% in ILC and 24% in DCIS (*p* < 0.0001). A total of 34% of IDC cases presented with high TrkA staining (staining intensity 2 and 3), whereas only 16% of ILC cases had strong TrkA staining. In contrast, all TrkA-positive DCIS presented with weak staining intensity (score 1) and no cases of normal breast tissue expressed TrkA. The most significant association of TrkA protein expression was found with HER2 expression. TrkA expression was found in 25% of HER2-negative tumours compared to 48% of HER2-positive tumours (*p* = 0.003). There were 41% of HER2-positive tumours presenting with high level of TrkA (scores 2 and 3) compared to only 20% of HER2-negative tumours. Log-linear modelling and subsequent analysis indicated that the odds of HER2 positivity increased by a factor of exp(0.56) = 1.75 (i.e., increased by 75%) in the presence of TrkA.

### 3.3. TrkA Is Expressed in Breast Cancer and Overexpressed in HER2-Positive Cell Lines

Western blot analysis was used to detect TrkA (total and phosphorylated) and HER2 protein expression across a panel of breast cancer cell lines (Figure 3). The data indicate that TrkA was detected as a 140 kDa band in all tested cell lines (Figure 3). In addition, a higher molecular weight form of TrkA (~180 kDa), probably reflecting TrkA N-glycosylation [33], was also detected in SK-BR-3 and BT-474 cell lines. Together the levels of TrkA appeared to be higher in three HER2-positive cell lines (SK-BR-3, BT-474 and MDA-MB-453) as compared to the other cell lines (HME, MCF-7, MDA-MB-231, 231-BR, MDA-MB-468, JIMT-1). The use of anti-phospho-TrkA antibodies demonstrated that only the 180 kDa form of TrkA is tyrosine phosphorylated in breast cancer cells, as no phospho-TrkA band could be detected at 140 kDa (Figure 3). The detection of HER2 protein was confirmed for the HER2-positive breast cancer cell lines and β-actin was used as the equal loading control for all Western blotting experiments.

### 3.4. Cytotoxic Effect of the Trk Inhibitor GNF-5837 in HER2-Positive Breast Cancer Cells

To define the impact of TrkA signaling in HER2-positive breast cancer cells, we treated SK-BR-3, BT-474, MDA-MB-453 and JIMT-1 cells with the Trk inhibitor GNF-5837 [34,35] and assessed drug sensitivity, impact on viability and downstream cellular signaling (Figure 4). The data show that HER2-positive breast cancer cells are sensitive to GNF-5837 and that viability is reduced in a dose-dependent manner for all cell lines: SK-BR-3 (IC_50_ = 2.59 µM, Figure 4A), BT-474 (IC_50_ = 4.66 µM, Figure 4B), MDA-MB-453 (IC_50_ = 4.69 µM, Figure 4C) and JIMT-1 (IC_50_ = 15.31 µM, Figure 4D). The human cervical cancer cell line C4-1 (TrkA negative) was used as a control and did not respond to treatment with GNF-5837 (data not shown). TrkA phosphorylation and various downstream tumourigenic and metastatic-related pathways such as AKT, Erk1/2 and SRC were also examined following treatment with 0–20 µM of GNF-5837 (Figure 4E). Phosphorylated TrkA (Tyr490) and phospho-AKT (Ser473) were markedly decreased in response to GNF-5837, in a dose-dependent manner, and common to all HER2-positive cell lines (Figure 4E). Downstream phospho-SRC (Tyr416) was reduced only in JIMT-1 cells and phospho-Erk1/2 (Thr202/Tyr204) was decreased in SK-BR-3 cells (Figure 4E), following treatment with GNF-5837. However, GNF-5837 did not increase the cytotoxic effect of Herceptin in any of the tested cell lines (Figure 4A–E). These results suggest that targeting TrkA signaling can decrease the viability of HER2-positive breast cancer cells but does not potentiate the impact of Herceptin in HER2-expressing breast cancer cell lines.

## 4. Discussion

This study has clarified the expression of TrkA in large and clinically diverse cohorts of breast cancer. In addition, we have highlighted a potential relationship between the tyrosine kinase receptors TrkA and HER2, demonstrating that the Trk inhibitor GNF-5837 can be used to specifically target TrkA signaling in HER2-positive breast cancer cell lines. The HER2-positive breast cancer cell lines chosen for functional analysis are diverse and include the anti-HER2-resistant JIMT-1 and MDA-MB-453, as well as SK-BR-3 and BT-474 which are sensitive to anti-HER2 treatment. All four HER2-positive cell lines have the expected *ERBB2*/HER2 amplification, while additional mutations are present in BT-474 (*RPS6KB1* and *ZNF217*), SK-BR-3 (*ZNF217*) and JIMT-1 (*JUN*, *CDK4*, *SLUG*, *GSTM1* and *CYP24*) [36,37,38].

Prior to immunohistochemical profiling and functional investigations, data mining of TrkA (*NTRK1*) gene expression was performed using the cBioPortal cancer genomics platform of the PAM50 dataset from The Cancer Genome Atlas (TCGA) database, which demonstrated that alterations are observed in a moderate proportion of breast tumours. TrkA (*NTRK1*) gene expression has already been described in a cohort of breast carcinomas, in which TrkA mRNA was found to be expressed in most tumours and there was a slight correlation with ER and PR [23]; however, this study only focused on the transcriptomic level, with neither HER2 nor the different molecular subtypes of breast tumours included in the analysis. Our analysis was performed on a larger and more diverse cohort of breast tumours (PAM50), including all molecular subtypes and HER2 status, however no statistically significant associations were found between TrkA (*NTRK1*) gene alterations and the clinicopathological parameters. It is now well established that the level of mRNA has a limited value for predicting the corresponding protein level. Initial studies in yeast have suggested a correlation of approximately 50% between mRNA and protein levels [39], but more recently in humans, multiple global transcriptomic and proteomic analyses have shown that only 30–60% of changes in protein levels are directly related to changes in mRNA [40,41]. Genetic mutations and alterations as well as a variety of different miRNA families have now been identified as critical regulators of protein synthesis and post-translational modifications, which can in turn underpin these observed discrepancies [41]. In addition, a recent proteogenomic investigation in colorectal cancer has revealed that mRNA abundance does not reliably predict protein abundance in tumours [42]. This emphasises the importance of analysing the protein level in order to define new biomarkers and therapeutic targets in oncology.

In developing neurons, NGF binds to and activates TrkA, which results in the stimulation of neuronal growth through a variety of well-established signaling pathways [7]. In the adult, TrkA also acts as a pain receptor in sensory neurons and contributes to the transmission of pain signals to the central nervous system [8]. Outside of neuronal growth and pain signaling, there is an increasing number of studies demonstrating that TrkA tumour-promoting signaling is present in several human cancers, including those of the breast [10], oral cavity [11], thyroid [13], cervix [43] and lung [14,15]. In our study, TrkA protein expression was found preferentially in IDC as compared to DCIS and ILC. IDC represent most breast cancers (~80%) and are generally more aggressive than DCIS and ILC, suggesting that TrkA overexpression is related to breast tumour aggressiveness. The tyrosine kinase receptor HER2 is a major clinical biomarker and therapeutic target in breast cancer. Approximately 20% of breast cancers express HER2 and the development and clinical use of a monoclonal antibody against HER2 (Trastuzumab/Herceptin or Lapatinib/Tykerb) has considerably improved survival outcomes for these patients. Nevertheless, resistance to anti-HER2 therapies occurs in 50% of patients with HER2-positive tumours [1] and the discovery of alternative therapeutic targets remains an essential objective for improving the management of HER2-positive tumours. Our findings demonstrate that TrkA protein expression is increased in HER2-positive breast cancers, which suggests TrkA as a complementary therapeutic target for these tumours, particularly those which are resistant to anti-HER2 therapies.

In vitro and animal model experiments have previously shown that TrkA overexpression enhances growth and invasion of breast cancer cells through the activation of Erk1/2 and PI3K-AKT mediated signaling pathways [10]. Our present findings show that TrkA and downstream AKT signaling modulate cell growth and viability and can be targeted in HER2-expressing breast cancer cells. Interestingly, an in vitro study has previously suggested a physical interaction between TrkA and HER2 in breast cancer cells [44]. The two receptors were co-immunoprecipitated and HER2 was necessary to obtain a stimulation of breast cancer cell growth, via downstream activation of Erk1/2. In addition, the recent demonstration that NT-3 (a TrkA ligand) can increase the expression of HER2 in breast cancer cells [45] provides further indication of a relationship between these two tyrosine kinase receptors in breast cancer. Importantly, several pharmacological inhibitors of Trk have been developed and are currently being tested on various cancers in a clinical setting [17,18,19]. The present study suggests that these Trk inhibitors, such as the GNF-5837 that we have used here, should also be evaluated for HER2-positive breast cancer, particularly for cases that are resistant to current therapeutic approaches, as they may represent future candidates of clinical importance.

## 5. Conclusions

In this study, we demonstrate an increased expression of the receptor tyrosine kinase TrkA in HER2-positive breast cancer. In addition, we found that the pharmacological targeting of TrkA signaling resulted in a decreased survival of HER2-positive breast cancer cells. Importantly, approximately 50% of HER2-positive breast cancers are resistant to anti-HER2-targeted therapies and our results suggest that TrkA could be used as an alternative therapeutic target for these tumours.

## Figures and Tables

**Figure 1 biomolecules-10-01329-f001:**
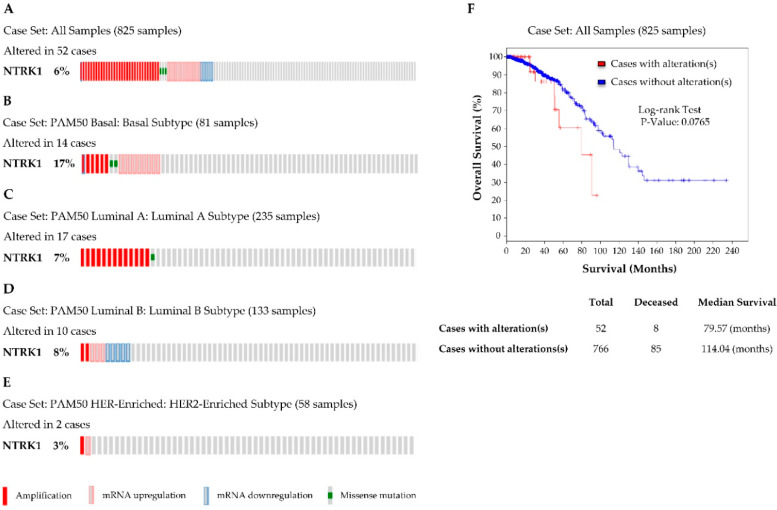
*NTRK1* (TrkA) gene expression in breast cancer patients from The Cancer Genome Atlas (TCGA) PAM50 dataset. The cBioPortal platform was used to data mine the TCGA PAM50 dataset. *NTRK1* (TrkA) gene expression, number and percentage of altered cases (amplification, mRNA upregulation, mRNA downregulation, missense mutations) are reported for (**A**) all breast tumours as well as for the following molecular subtypes of breast cancer: (**B**) basal, (**C**) luminal A, (**D**) luminal B and (**E**) HER2 enriched. (**F**) Overall patient survival for all cases of invasive breast carcinomas (*n* = 825), with *NTRK1* (TrkA)-altered cases in red and *NTRK1* (TrkA)-unaltered cases in blue. The number of total and deceased cases are reported in addition to medium survival (months).

**Figure 2 biomolecules-10-01329-f002:**
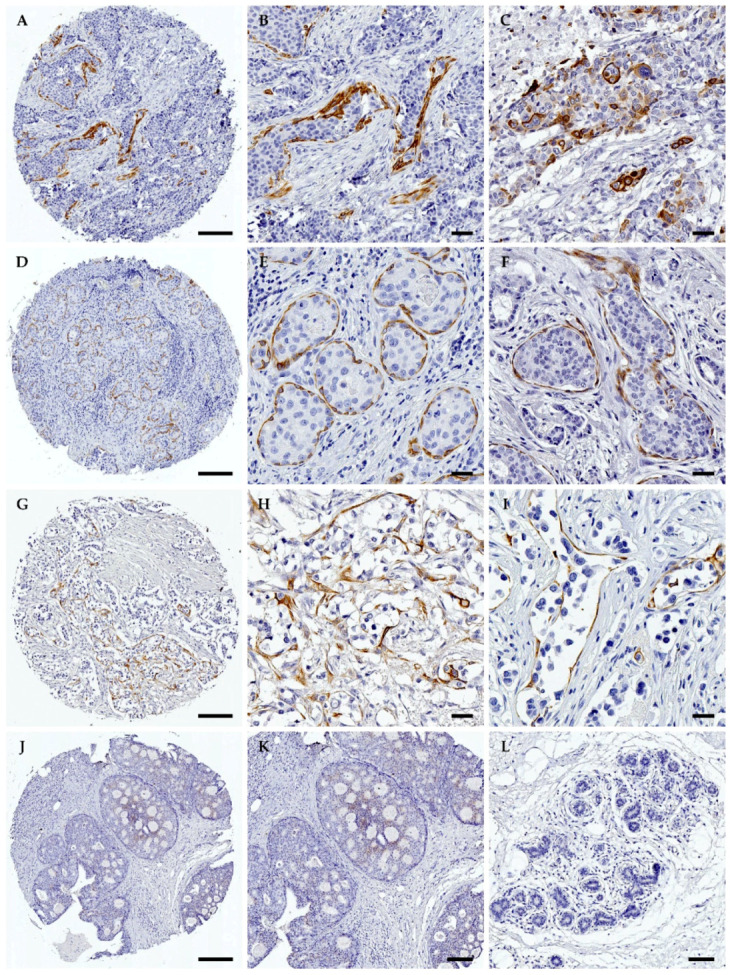
Immunohistochemical detection of TrkA in human breast cancers. The protein expression of TrkA was assessed by immunohistochemistry in a series of invasive breast cancers and ductal carcinomas in situ (DCIS). TrkA immunolabelling was observed in 39% of invasive ductal carcinomas (IDC), 20% of invasive lobular carcinomas (ILC) and 24% of DCIS, with staining mostly concentrated in the cancer epithelial cells and myoepithelium. Representative images of TrkA immunolabelling are shown. (**A**,**D**) IDC entire core. (**B**,**E**) Enlargement of areas in A and D, respectively. (**C**,**F**) IDC high magnification. For IDC, TrkA labelling was observed in cancer epithelial cells and myoepithelial cells. (**G**) ILC entire core. (**H**,**I**) Higher magnification of areas found in G. For ILC, some myoepithelial and cancer epithelial cells were positive for TrkA. (**J**) DCIS entire core. (**K**) Higher magnification of area in J. TrkA immunolabelling was rarely observed in DCIS. (**L**) Normal breast tissue. TrkA immunolabelling was not observed in normal breast tissues. Quantification of TrkA immunolabelling is reported in Table 1. *n* = 158 IDC, 158 ILC and 50 DCIS. Scale bars: 50 µm. Original magnification: x50 and x400 for entire cores and higher magnification regions, respectively.

**Figure 3 biomolecules-10-01329-f003:**
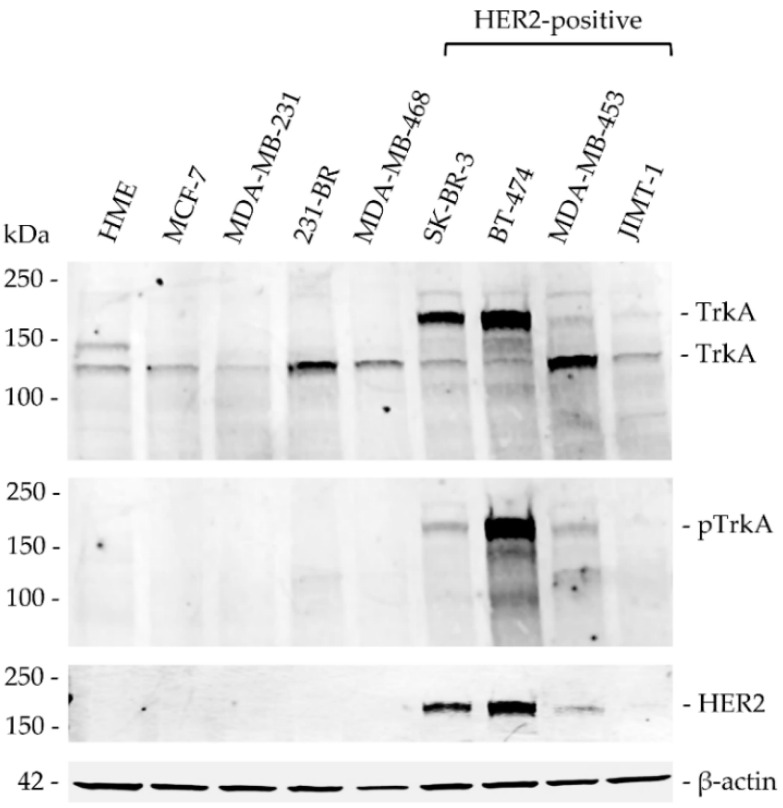
TrkA, phospho-TrkA and HER2 protein expression in a panel of breast cancer cell lines. Western blotting for TrkA was performed with cellular proteins extracted from HME (non-tumourigenic human mammary epithelial) cells and the following breast cancer cell lines: MCF-7 (luminal A), MDA-MB-231 (triple-negative) and 231-BR (brain metastatic derivative), MDA-MB-468 (triple-negative); as well as the following the HER2-positive cell lines: SK-BR-3, BT-474, MDA-MB-453 and JIMT-1. TrkA was detected as a 140 kDa band in all cell lines. In addition, a second band at 180 kDa was observed in SK-BR-3 and BT-474 cell lines. Another band at 150 kDa was detected only in HME cells. The intensity of TrkA immunoreactive bands was higher in the HER2-positive cell lines SK-BR-3, BT-474 and MDA-MB-453. Phospho-TrkA (p-TrkA) immunoreactive bands were observed at 180 kDa in SK-BR-3, BT-474 and MDA-MB-453 cell lines. HER2 protein expression was confirmed across all HER2-positive breast cancer cell lines and β-actin protein expression was used as the equal loading control.

**Figure 4 biomolecules-10-01329-f004:**
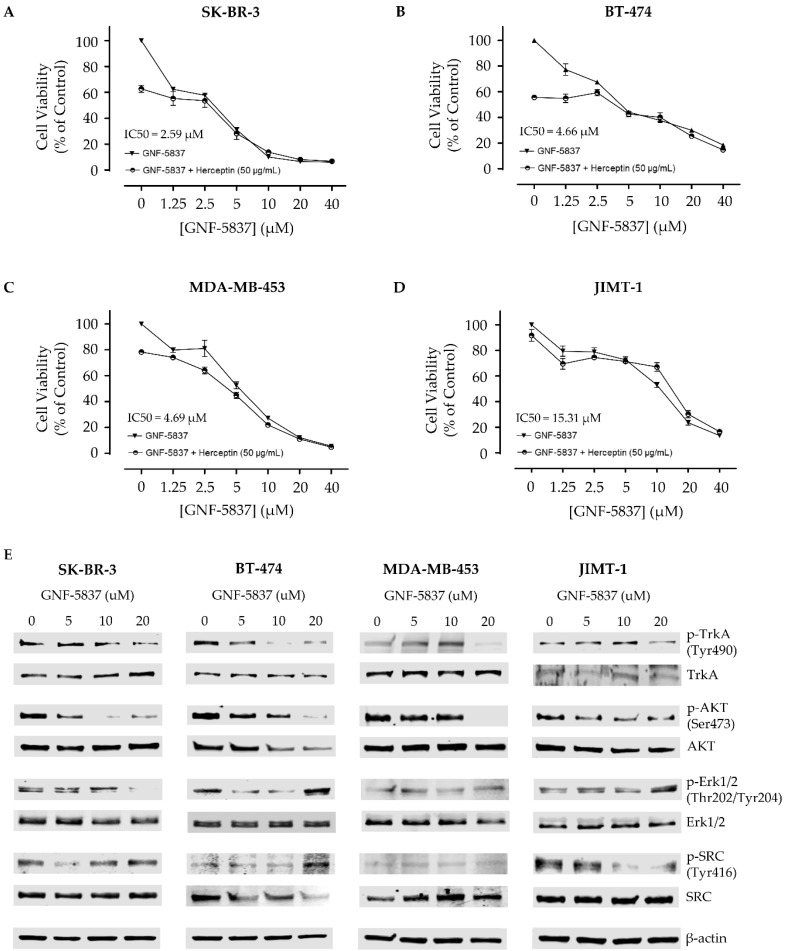
Impact of the Trk inhibitor GNF-5837 on HER2-positive breast cancer cell lines. HER2-positive breast cancer cells were treated with 0–40 µM grade concentrations of the Trk inhibitor GNF-5837, with and without 50 µg/mL Herceptin, in media containing 10% FBS for 48 h and a dose–response curve and associated IC_50_ values were generated for (**A**) SK-BR-3, (**B**) BT-474, (**C**) MDA-MB-453 and (**D**) JIMT-1. Data are presented as the mean ± SD of samples from three independent experiments. (**E**) Effect of 0–20 µM GNF-5837 on TrkA phosphorylation and downstream cellular signaling. Western blot analysis was performed 48 h post-treatment with GNF-5837. The level of phospho-TrkA (p-TrkA) and phospho-AKT (p-AKT, Ser473) both decreased in a dose-dependent manner in all HER2-positive breast cancer cell lines. The 0.1% DMSO was used as the negative control treatment for the viability assay and β-actin protein expression was used as an equal loading control in Western blotting.

**Table 1 biomolecules-10-01329-t001:** Association between TrkA expression and clinicopathological parameters in human breast carcinomas.

	TrkA Intensity				
Parameter	0	1	2	3	*p-*Value
All Cases (*n* = 366)	260 (71%)	27 (7%)	48 (13%)	31 (9%)	
***Pathological Subtype***					**<0.0001 ^a^**
DCIS (*n* = 50)	38 (76%)	12 (24%)	0 (0%)	0 (0%)	
ILC (*n* = 158)	126 (80%)	7 (4%)	19 (12%)	6 (4%)	
IDC (*n* = 158)	96 (61%)	8 (5%)	29 (18%)	25 (16%)	
**Clinical Parameters in Invasive Carcinomas**					
***Patient Age*** *(**Years**)*					0.7195
≤50 (*n* = 184)	130 (71%)	7 (4%)	30 (16%)	17 (9%)	
>50 (*n* = 132)	92 (70%)	8 (6%)	18 (14%)	14 (10%)	
***Lymph Node Status*** *(**N**)*					0.0696
Negative (*n* = 163)	121 (74%)	9 (6%)	21 (13%)	12 (7%)	
Positive (*n* = 133)	82 (62%)	6 (5%)	27 (20%)	18 (13%)	
**HER2**					**0.0027 ^a^**
HER2 negative (*n* = 250)	188 (75%)	10 (4%)	31 (12%)	21 (9%)	
HER2 positive (*n* = 66)	34 (51%)	5 (8%)	17 (26%)	10 (15%)	
***Estrogen Receptor***					0.2267
ER negative (*n* = 181)	129 (71%)	5 (3%)	27 (15%)	20 (11%)	
ER positive (*n* = 135)	93 (69%)	10 (7%)	21 (16%)	11 (8%)	
***Progesterone Receptor***					**0.0261 ^a^**
PR negative (*n* = 207)	152 (73%)	6 (3%)	26 (13%)	23 (11%)	
PR positive (*n* = 109)	70 (64%)	9 (8%)	22 (20%)	8 (8%)	
***Tumour Size*** *(**T**)*					0.6360
1 + 2 (*n* = 238)	166 (70%)	14 (6%)	36 (15%)	22 (9%)	
3 + 4 (*n* = 58)	38 (65%)	1 (2%)	11 (19%)	8 (14%)	

TrkA = tyrosine kinase receptor A; DCIS = ductal carcinoma in situ; ILC = invasive lobular carcinomas; IDC = invasive ductal carcinomas; HER2 = human EGF receptor 2; ER = estrogen receptor; PR = progesterone receptor; TNBC = triple-negative breast cancer (ER-/PR-/HER2-). ^a^ Statistically significant *p*-values (*p* < 0.05 using chi-square test).

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
