# Peer review of "The Receptor Tyrosine Kinase TrkA Is Increased and Targetable in HER2-Positive Breast Cancer"

_biomolecules, 2020, doi:10.3390/biom10091329_

Round 1

Reviewer 1 Report

The manuscript by Griffin and colleagues aims to identify the IHC expression pattern of TrkA, a tyrosine kinase receptor, in various breast cancer subtypes and histopathological profiles.  They found that increased TrkA expression correlates with invasive ductal carcinoma compared to invasive lobular carcinoma and the less invasive DCIS. Additionally, they found that TrkA expression positively correlated with HER2 expression, as more TrkA expression was found in HER2+ breast tumors. While the study is a well thought out study, some things need to be addressed before it can be published:

Major:

  1. I find it interesting that HER2+ breast cancer has the lowest percentage of mutations in the TrkA gene, but TrkA expression positively correlates with HER2. Can the authors comment on this?
  2. For Figure 4 A-D, it would be nice to see the effects of the TRKA inhibitor on a cell line that does not express high levels to rule out a general effect.
  3. Why did the authors choose to use GNF-5837 and not a more clinically relevant drug that is currently in clinical trials?

Minor:

  1. For Figure 1F, is the overall survival of patients with an alteration a combination of both up and down regulation of the mRNA transcript? It is interesting that the combination of all those different mutations has such a significant effect on the patient survival. Would this suggest that any type of alteration to mRNA expression has a dramatic effect on patient viability (i.e. an increase or decrease is bad for the patient, and a tight regulation of this gene is necessary to keep it within the appropriate bounds)?
  2. In Figure 4E MDA-MB-453 is misspelled.
  3. For Figure 4E, it is difficult to point to something that is common to any of the cell lines other than a decrease in TrkA and pTrkA. This suggests that effects on drug sensitivity may not be through these mechanisms.
  4. Lines 343-345 the authors state that ”…TrkA as a complementary therapeutic target for these tumours, particularly those which are resistant to anti-HER2 therapies”. The authors could test this with dual treatment of the cell lines using the drug curves already done in Figure 4.

Author Response

Reviewer 1

Major:

  1. I find it interesting that HER2+ breast cancer has the lowest percentage of mutations in the TrkA gene, but TrkA expression positively correlates with HER2. Can the authors comment on this?

We agree with the reviewer that this is indeed an interesting observation. We eluded to a possible explanation in the discussion (page 10, lines 317-327) and emphasised the study by Zhang et al. (ref. 39, Nature, 2014) which highlighted that gene alterations and/or mRNA abundance does not reliably predict the abundance of protein in human tumours.

  1. For Figure 4 A-D, it would be nice to see the effects of the TRKA inhibitor on a cell line that does not express high levels to rule out a general effect.

We have previously tested the effect of the TrkA inhibitor GNF-5837 on cell lines (human cervical cancer cell line C4-1) that do not express TrkA. We observe that there is no significant effect on cell viability when using doses identical to those reported in this study. We have now indicated this point in the text (page 3, lines 119-120 and page 8, lines 285-287).

  1. Why did the authors choose to use GNF-5837 and not a more clinically relevant drug that is currently in clinical trials?

We chose to investigate the effect of the drug GNF-5837 on TrkA signalling in HER2-positive cell lines as the research team has previously published findings using this drug, for which we observe a highly specific and sensitive response in the pTrkA signalling pathway.  GNF-5837 is also a selective, orally bioavailable drug that does not need to be administered systemically. This may therefore be used in follow up studies involving animal models of HER2-positive breast cancer.

Minor:

  1. For Figure 1F, is the overall survival of patients with an alteration a combination of both up and down regulation of the mRNA transcript? It is interesting that the combination of all those different mutations has such a significant effect on the patient survival. Would this suggest that any type of alteration to mRNA expression has a dramatic effect on patient viability (i.e. an increase or decrease is bad for the patient, and a tight regulation of this gene is necessary to keep it within the appropriate bounds)?

We thank the reviewer for raising this interesting point. Indeed, the overall survival of patients includes alterations with a combination of both up and down regulations of the mRNA transcript. Taking into consideration that NTRK1/TrkA is an established oncogene and that most alterations observed in our analyses are amplifications and/or mRNA upregulations, it suggests that these are most likely responsible for changes to patient viability/survival. We have now added this additional point into the text (page 4, lines 184-186).

  1. In Figure 4E MDA-MB-453 is misspelled.

MDA-MB-453 is now correctly spelled in Figure 4E (page 9).

  1. For Figure 4E, it is difficult to point to something that is common to any of the cell lines other than a decrease in TrkA and pTrkA. This suggests that effects on drug sensitivity may not be through these mechanisms.

We agree with the reviewer’s observation in part. Indeed, pTrkA is decreased following treatment with GNF-5837, which indicates the specificity of the drug. We would however like to take this opportunity to point out the response within the AKT signalling pathway, specifically the reduction in pAKT (Ser473), which was common across all tested HER2+ breast cancer cell lines. For SK-BR-3, BT-474 and JIMT-1, the reduction in pAKT (Ser473) is related to the concentration of GNF-5837 (dose-dependent), therefore highlighting drug sensitivity and ruling out a general effect. We have now emphasised this point in the text (page 8, line 291)

  1. Lines 343-345 the authors state that ”…TrkA as a complementary therapeutic target for these tumours, particularly those which are resistant to anti-HER2 therapies”. The authors could test this with dual treatment of the cell lines using the drug curves already done in Figure 4.

MDA-MB-453 and JIMT-1 HER2+ breast cancer cell lines are known to be resistant to anti-HER2 treatment and we also report a limited response to Herceptin (~ 80-90% viability) as compared to the more sensitive SK-BR-3 and BT-474 (50-60% viability) cell lines. This highlights the potential utility of GNF-5837 and/or other inhibitors of TrkA phosphorylation/signalling as a complimentary and/or second-line therapy, as we demonstrate a reduction in cell viability across both anti-HER2 resistant and anti-HER2 sensitive cell lines.

Reviewer 2 Report

Having read the manuscript "The receptor tyrosine kinase TrkA is increased and targetable in HER2-positive breast cancer" I have the following comments:

  1. What is the genetic background of the breast cancer cell lines used in this study?  What mutations are observed in JIMT-1 cells compared to the other cell lines tested especially the other HER2-positive cell lines.  Can the authors please add this information
  2. The citation for reference number 40 is incomplete.  Can the authors also standardise their reference list and either list the abbreviated or full names of the journals in this list.

Author Response

Reviewer 2

  1. What is the genetic background of the breast cancer cell lines used in this study? What mutations are observed in JIMT-1 cells compared to the other cell lines tested especially the other HER2-positive cell lines.  Can the authors please add this information?

The genetic background of the HER2-positive breast cancer cell lines used in this study, including three references (36-38), has been added to the text (page 10, lines 314-319).

  1. The citation for reference number 40 is incomplete. Can the authors also standardise their reference list and either list the abbreviated or full names of the journals in this list.

We thank the reviewer for picking up on the incompleteness of reference number 40, which is now reference 43 in the updated manuscript. All references have now been checked and updated using the MDPI endnote referencing style and cross-checked with the journal guidelines.